# A Novel Active Device for Shoulder Rotation Based on Force Control

**DOI:** 10.3390/s23136158

**Published:** 2023-07-05

**Authors:** Isabel M. Alguacil-Diego, Alicia Cuesta-Gómez, David Pont, Juan Carrillo, Paul Espinosa, Miguel A. Sánchez-Urán, Manuel Ferre

**Affiliations:** 1Physiotherapy, Occupational Therapy, Physical Medicine and Rehabilitation Department, Universidad Rey Juan Carlos, Campus de Alcorcón, Av. de Atenas s/n, Alcorcón, 28922 Madrid, Spain; isabel.alguacil@urjc.es; 2Centre for Automation and Robotics (CAR) UPM-CSIC, Universidad Politécnica de Madrid, C/. José Gutierrez Abascal, 2, 28006 Madrid, Spain; david.pont@upm.es (D.P.); juan.carrillo.rios@gmail.com (J.C.); paul.espinosa.peralta@alumnos.upm.es (P.E.); miguelangel.sanchezuran@upm.es (M.A.S.-U.); m.ferre@upm.es (M.F.)

**Keywords:** force control, haptic, upper limb, rehabilitation, shoulder movements, medical rehabilitation

## Abstract

This article describes a one-degree-of-freedom haptic device that can be applied to perform three different exercises for shoulder rehabilitation. The device is based on a force control architecture and an adaptive speed PI controller. It is a portable equipment that is easy to use for any patient, and was optimized for rehabilitating external rotation movements of the shoulder in patients in whom this was limited by muscle–skeletal injuries. The sample consisted of 12 shoulder rehabilitation sessions with different shoulder pathologies that limited their range of shoulder mobility. The mean and standard deviations of the external rotation of shoulder were 42.91 ± 4.53° for the pre-intervention measurements and 53.88 ± 4.26° for the post-intervention measurement. In addition, patients reported high levels of acceptance of the device. Scores on the SUS questionnaire ranged from 65 to 97.5, with an average score of 82.70 ± 9.21, indicating a high degree of acceptance. The preliminary results suggest that the use of this device and the incorporation of such equipment into rehabilitation services could be of great help for patients in their rehabilitation process and for physiotherapists in applying their therapies.

## 1. Introduction

Shoulder rotator cuff injuries include tendinopathies and partial or complete tendon ruptures, mostly related to the supraspinatus tendon [1]. Exercise is a key component of treatment for shoulder soft tissue injuries of the shoulder. Evidence recommends exercise-based interventions to reduce pain and increase shoulder function. Evidence also suggests that progressive shoulder strengthening and stretching are effective in the management of rotator cuff injuries [2]. Exercises should be introduced as early as possible. However, in clinical practice, the actual referral rate to physical therapists is low due to the lack of access to publicly funded physical therapy in health care systems [3]. Waiting for outpatient physiotherapy services has been shown to have mixed results in clinical and health system outcomes. The review by Deslauriers et al. [4] suggests the possible detrimental effects of waiting on pain, disability, quality of life, and psychological symptoms in people with shoulder disorders. There is also evidence of higher healthcare utilization and costs for patients who wait longer before physiotherapy services [4]. The effects of waiting caused by lack of access to physiotherapy services could be mitigated by the implementation of robotic devices with which more patients can be treated at the same time with a lower number of physiotherapists. Rehabilitation robots can significantly reduce the burden of therapists by providing repetitive and precise therapy to people with upper limb impairments over a long period of time [5].

Most upper extremity devices for rehabilitation available on the market are designed to improve flexion and abduction, and not external rotation, which is a fundamental movement in the performance of activities of daily living. Rotation movements are fundamental for performing activities below the horizontal and to execute coordinated movements with the hand for an individual to locate themselves in space [6]. In addition, most studies have been carried out in stroke patients [7,8]. This equipment is based on exoskeletons [9] or haptic devices [10]. The exoskeletons typically cover the arm and forearm and are linked to the trunk using flexible components. Meanwhile, haptic devices are attached to the hand to guide the movement while exerting force.

There is great interest in developing more specific equipment for rehabilitation [11]. This equipment is focused specifically on rehabilitation and solve the constraints of generic exoskeletons, particularly, the difficulties of supporting complex movements of joints such as the shoulder. The rigid exoskeletons use special mechanisms that increase its complexity. At the end, the goal is to adjust the actuated degree of freedom of the exoskeleton to the desired rotation plane of the shoulder/arm for each specific rehabilitation treatment. In some cases, textile or other flexible materials are used to avoid movement constraints.

This article describes a one-degree-of-freedom haptic device that can be applied to perform three different exercises for shoulder rehabilitation. This device implements force control behavior. This force control entails resistance to the patient’s shoulder rotation that is customized according to the patient’s status. Therefore, the same equipment can be adapted to the patient’s evolution and all session activities can be recorded for clinician studies. A portable and compact equipment has been designed for shoulder rehabilitation treatments.

## 2. Materials and Methods

For the present study, a pre- and post-treatment study was performed for an intervention session using a one-degree-of-freedom haptic device to perform three different exercises for shoulder rehabilitation.

### 2.1. Device

The rehabilitation device consists of the following components: the actuator, transmission components, force sensor, handle, and controller. The architecture used is shown in Figure 1. The actuator is based on a DC motor; the Maxon DCX22S has been selected for this application. Its rated operation is at 38 VDC, 5.9 Nm, and 10.5 revolutions per minute (RPM). This motor generates the required force that is applied to the patient. The controller calculates the reference value according to the desired force and information provided by sensors; then, the driver (H-bridge max14870) sets the power to the motor. The controller is run on a Texas Instruments board, specifically, the LAUNCHXL-F28379D model. This board contains a 32-bit dual-core TMS320F2-8379D microcontroller, which runs at 200 MHz. The controller closes the force control loop at 1 kHz. The transmission is made up of a cable and a pulley that transmit the actuator torque to the patient; the ratio between the motor and the pulley is 100:1 mm. A force sensor is located at the end of the cable and is also linked to the handle. The force sensor is implemented by a load cell manufactured by Futek, which is used to measure the force reflected by the motor. The sensor model used in this work is LSB205 connected to the Analogue Strain Gauge Signal Conditioner IAA100; it offers a load measurement range from 0 to 11 Kg, corresponding to 0 to 5 DC volts.

The patient’s hand is positioned inside the handle, which is made of a flexible textile for a comfortable grip. Figure 2 shows an image with all the components. The patients sit in a chair, and the described components are mounted in an aluminum structure that provides a comfortable workspace.

### 2.2. Control Algorithm Design

In this work, a force controller is proposed based on the cable tension exerted by the patient and an adaptive speed PI controller. The force sensor provides a range of detection from 0 N to 111 N; therefore, the maximum configurable payload of the prototype is 11.32 Kg. The proportional gain K_p_ is modified as a function of the ratio between the force error and the reference; the proposed values are shown in Figure 3. The integral gain K_i_ was set at a value of 2. K_p_ and K_i_ were tuned experimentally.

### 2.3. Controller Performance

Some trials were performed in the laboratory to properly tune the controllers. Figure 4 shows the performance of the force exerted by the motor when applied to the reference. As we can see, the information provided by the force sensor oscillates around the reference. This oscillation is due to the interaction with the patient; the different movement speed is directly transformed to an incremental variation of the reflected force. This is the usual behavior of a force control loop that maintains a reflected force.

### 2.4. Participants 

The inclusion criteria established for the selection of patients were as follows: upper limb musculoskeletal injury, no pathology for which rehabilitation treatment is contraindicated, and over 18 years of age. Patients who met any of the following exclusion criteria were not admitted to the study: cognitive impairment that prevented the understanding of simple commands, neurological lesion affecting the upper extremity, dermatological lesion of the upper extremity preventing the use of device material, and/or any other type of injury impeding the use of the device.

This protocol was approved by the Research Ethics Committee of the Rey Juan Carlos University Institute. The ethical principles for medical research in humans from the Declaration of Helsinki were followed. All subjects signed the informed consent form prior to participation. Trials were carried out at the Getafe Clinical Centre (Madrid, Spain). 

### 2.5. Intervention

The clinician adjusted the controller parameters for each patient according to the predefined exercise (in this case, the shoulder rotation movement). These parameters define the shoulder rotation range and the stress of the reflected force on the patient, which is in an interval of 5 N to 15 N. Furthermore, it was verified that the pulley was in the same plane as the patient’s forearm. This alignment makes movements more comfortable as the forces directly oppose shoulder rotation with no movement at the elbow.

Each session lasted 15 min and consisted of repeating a cycle of external rotation and internal rotation 30 times, returning to the starting position. These cycles were carried out with the patient sitting down, in a chair with a backrest, and with their feet on the floor. Initially, the arm was positioned close to the body with a 90° elbow flexion, and from this position, the rotation movements were performed. Figure 5 shows a patient during a session. 

To perform the rotation movements, the patient started the external rotation movement from the initial position up to the maximum rotation they were able to achieve. In this half-cycle, the patient exerted a force greater than that generated by the motor. In the second half of the cycle, when the patient returned to the initial position, performing an internal rotation; this movement was in the opposite direction and the force applied by the motor was greater than that exerted by the patient. Figure 6 shows the limits of the cyclical movement. The force reflected by the actuator was fixed throughout the cycle and measured by the load cell. This load cell was located close to the patient’s hand to obtain an accurate estimate of the force reflected on the patient; otherwise, the forces caused by friction might affect the information processed by the controller.

### 2.6. Outcome Measures

The range of motion (ROM) of external rotation was measured using a double-armed 360° goniometer. The patients were in a supine position with the humerus abducted at 90° and the elbow flexed at 90°. Measurements were performed twice and averaged for further analysis [12]. 

The system usability scale (SUS) was used to assess patient satisfaction with the device. This questionnaire was developed by Brooke [13] as a usability tool and has been widely used in the evaluation of a variety of systems. The SUS provides a quick and reliable tool for measuring the usability of a device. It is a simple and short questionnaire to answer; a final score is provided with an interpretation based on a well-established reference standard. It has excellent reliability (0.85). It consists of a 10-item questionnaire with five response options, from strongly agree (score of 5) to strongly disagree (score of 1). There are five positive and five negative statements, which are presented in alternatingly. The odd-numbered questions (Q1 “I think I would use this device frequently’, Q3 “I think the device was easy to use’, Q5 “The functions of this device are well integrated’, Q7 “I imagine that most people would learn to use this device very quickly’, and Q9 “I feel safe using this device’) are positive questions, and the recorded scores are the original scores subtracted by 1. The even-numbered questions (Q2 “I find this device unnecessarily complex’, Q4 “I think I would need help from a person with technical knowledge to be able to use this device’, Q6 “I think the device has a lack of consistency’, Q8 “I find the device very difficult to use’, and Q10 “I needed to learn many things before able to use this device”) are negative questions with recorded scores being subtracted from 5. Once the results for the ten questions are treated, the score of each question is added, and the result is multiplied by a factor of 2.5. The SUS score, therefore, ranges from 0 to 100, where a higher score means better usability, with a threshold of 68 to establish the usability of the device [14].

### 2.7. Statistical Analysis

Statistical analysis was performed using the SSPS statistical software system (SSPS Inc., Chicago, IL, USA; version 27.0). A descriptive analysis of all the variables was performed. The results were expressed as average and standard deviation and median and interquartile range. The normal distribution of the variables was verified using the Shapiro–Wilk test. The hypothesis that the variables did not have a normal distribution was accepted, given the results of this test, the verification of each variable’s histograms, and the sample size. We used the Wilcoxon test, a nonparametric test for related samples. Statistical analysis was performed with a confidence level of 95%, and therefore significant values were those whose *p* was < 0.05.

## 3. Results

The sample consisted of 12 shoulder rehabilitation sessions with 11 patients (n = 11) (5 men and 6 women) with different shoulder pathologies that limited their range of shoulder mobility. In one of the patients, both shoulders were treated. One patient performed two sessions for the rehabilitation of both shoulders, while the rest performed only one rehabilitation session. The ages ranged from 35 to 66 years (54.92 ± 3.16 years). Table 1 shows the data of the patients who participated in the testing of the device.

The analysis shows statistically significant changes for the ROM of external shoulder rotations. The mean and standard deviations of the external rotation of shoulder were 42.91 ± 4.53° for the pre-intervention measurements and 53.88 ± 4.26° for the post-intervention measurements. Figure 7 and Table 2 show the data and range of movement of the patients during the rehabilitation sessions.

The patient scores on the SUS questionnaire ranged from 65 to 97.5, with an average score of 82.70 ± 9.21, suggesting a high degree of acceptance. These scores are higher than 68, indicating that this device can be rated “excellent” in the acceptability range. Table 3 shows the means of the scores obtained for each question. Questions Q1, 3, 5, 7, and 9 were positive questions and the averages obtained in these questions were above 4. It is worth noting that Q1 “I think I would use this device frequently” presented an average score of 4.75 ± 0.59 and Q9 “I feel safe using this device” showed an average score of 4.91 ± 0.27. Meanwhile, questions Q2, 4, 6, 8, and 10 were negative questions. The mean scores in these questions were 1, except in Q2 “I find this device unnecessarily complex” which had a mean score of 2.5 ± 1.55 and Q4 “I think I would need help from a person with technical knowledge to be able to use this device” with a mean score of 2.91 ± 1.49.

## 4. Discussion

The present study showed the use of an active device for shoulder rehabilitation based on a force control architecture. This preliminary study is based on a single treatment session to treat the external rotation movements of the shoulder in the patients in whom this was limited by musculoskeletal injuries. It also described the patient’s acceptance of the device, and the results show a high level of approval from the patients after the first trial.

It has been demonstrated that robotic and haptic technologies, such as force control and real-time signal processing, have an effective contribution in upper limb rehabilitation. There are many examples of complex exoskeletons [15,16] and robot-like equipment applied in upper limb rehabilitation [17]. Physical therapy can be a long, lengthy, and costly process, which can lead to loss of interest in individuals undergoing therapy. Reducing recovery time is a potential solution that may motivate individuals to participate and continue physical therapy. Rehabilitation with an exoskeleton can reach more patients while reducing physician interactions and overall rehabilitation costs [18].

This equipment can perform a wide variety of movements but is specifically programmed for executing rehabilitation tasks. It is also necessary to design the proper patient interface to fix or attach the patient’s upper limb to the robot structure. These kinds of solutions have the advantage of being able to adapt to different rehabilitation activities. Cost-effective automation devices, such as exoskeletons, are critical to the success of these interventions and rehabilitation programs. 

With a single treatment session, this active device for shoulder rehabilitation based on force control was found to significantly increase the external rotation movement of the shoulder in the patients who used it. The pre-intervention mean and standard deviations of the ROM of the external rotation of the shoulder were 42.91 ± 4.53°, and 53.88 ± 4.26° in the post-intervention measurements. In addition, patients reported high levels of acceptance of the device; their scores on the SUS questionnaire ranged from 65 to 97.5, with an average score of 82.70 ± 9.21, indicating a high degree of acceptance. These very promising results were obtained with minimal hardware and a user-friendly interaction with the patient. 

This device is in line with a new generation of compact and simple rehabilitation equipment that is portable and easy to use for the patient [19], in contrast to the classical robotic solutions mentioned above. The main advantage of this kind of solution is the simplicity and cost-effective aspects. Only one degree of freedom is enough to perform shoulder rehabilitation, and this equipment can be adapted for horizontal and vertical rotations. Moreover, the reduced size of the device allows it to be easily transported to the rehabilitation workspace, which can be located in a clinic or at the patient’s home.

This prototype is currently being improved in order to provide more functionalities to clinicians for performing new rehabilitation exercises and software is also being developed to better characterize the rehabilitation sessions. 

## 5. Study Limitations

This study has several limitations. First, the small sample size cannot allow extrapolation of the results. Second, the ROM was evaluated with a manual goniometer. Futures studies are necessary using objective measures such as using a electrogoniometer or 3D motion analysis. Third, the sample was heterogeneous in respect to the injury and age. It is necessary to take this into account. Finally, the lack of follow-up does not allow us to conclude if the ROM improvement lasted over time. 

## 6. Conclusions

In this paper, a compact and cost-effective prototype for shoulder rehabilitation was described. The most important features are the simplicity of the hardware and its effectiveness in rehabilitation exercises. The equipment is designed to repeat a shoulder rotation while reflecting a force on the patient’s hand. This exercise allows muscle activation, while also increasing the range of shoulder movement. 

This device allows patients to perform exercises for rehabilitation sessions that are an ideal complement to clinician manipulation. The preliminary results suggest that the use of this device and the incorporation of such equipment into rehabilitation services could be of great help for patients in their rehabilitation process and for physiotherapists in applying their therapies.

## Figures and Tables

**Figure 1 sensors-23-06158-f001:**
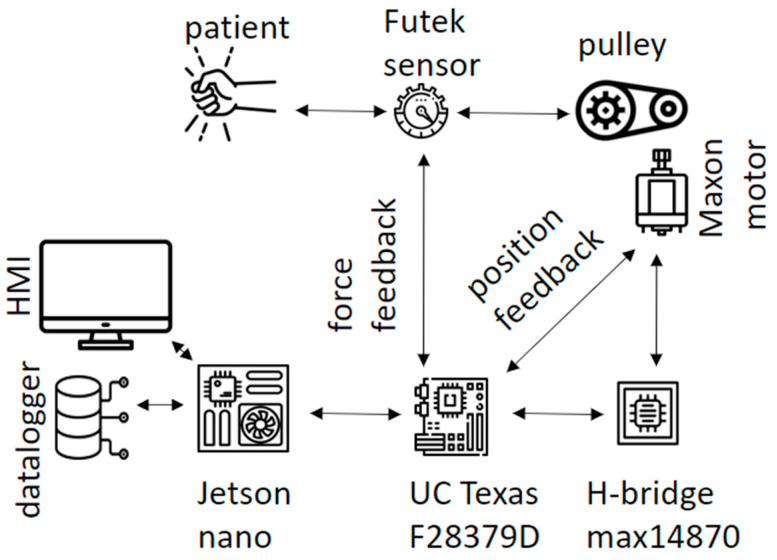
Proposed general architecture. The prototype consists of two electronic cards. the Jetson nano sends the setpoint parameters selected in the interface to the microcontroller (UC) Texas F28379D, and stores the values generated by the controller and the sensors. The H-bride transforms the PWM signal generated by the UC into a voltage to move the motor. The interaction between the patient and the motor is detected by the Futek load cell sensor.

**Figure 2 sensors-23-06158-f002:**
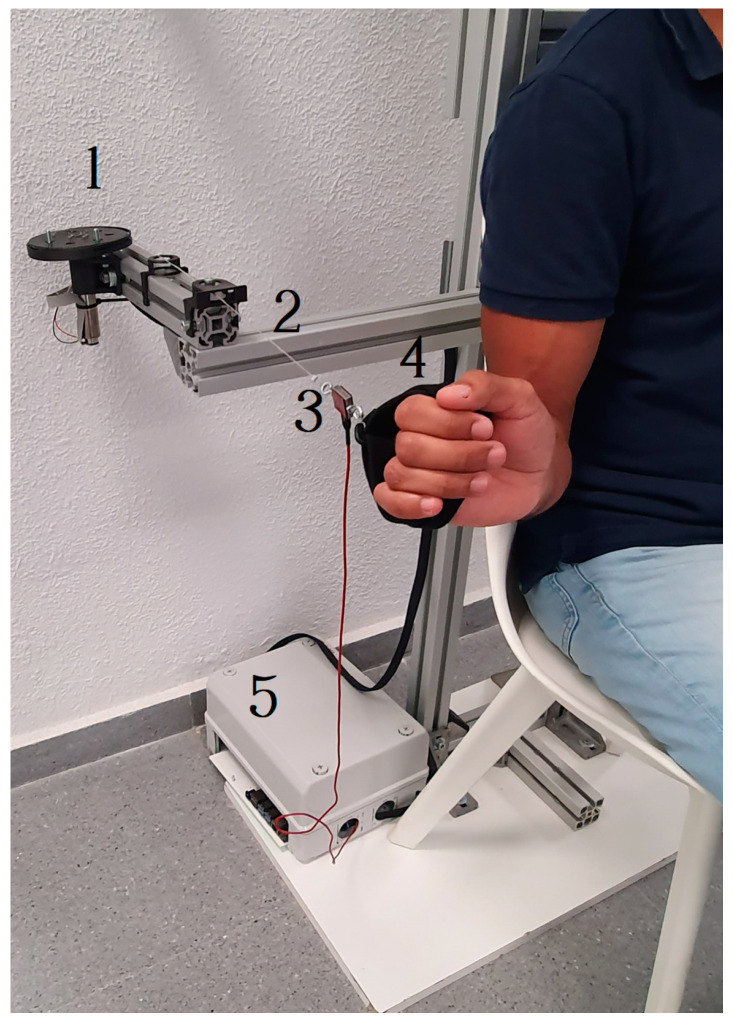
Main components of the rehabilitation device: actuator (1), transmission components (2), force sensor (3), handle (4), and controller (5).

**Figure 3 sensors-23-06158-f003:**
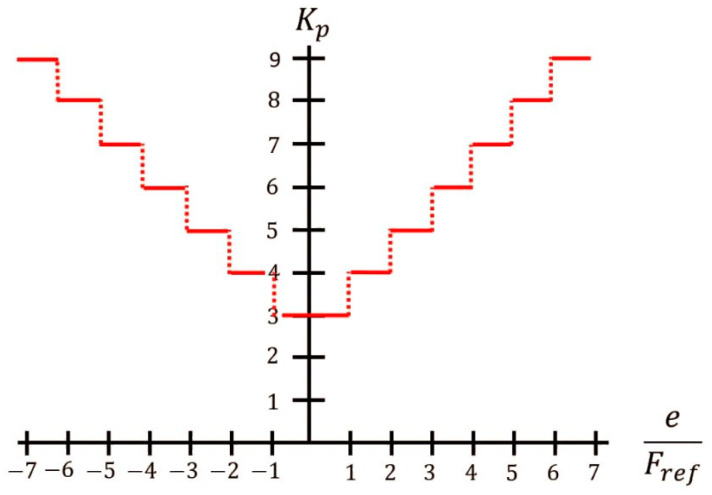
Value of the gain K_p_ according to the ratio between the force error and the reference. When the ratio increases, a higher gain is applied. The minimum value is adjusted for the recovering cable.

**Figure 4 sensors-23-06158-f004:**
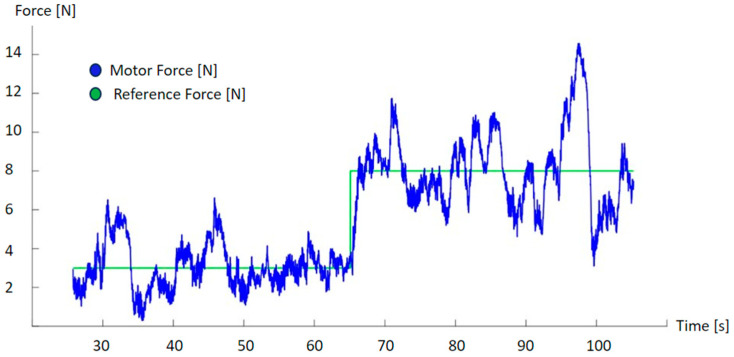
Evolution of the force exerted by the motor according to the force reference. Oscillations over the reference are due to the interaction with the patient to account for differences in resistance during the movement.

**Figure 5 sensors-23-06158-f005:**
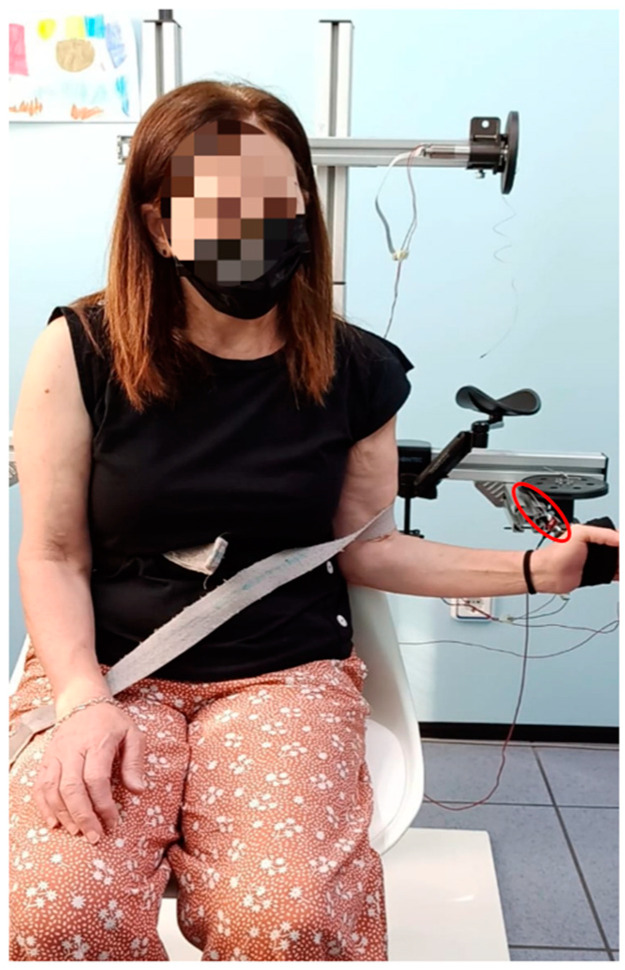
Patient performing shoulder rotation in the horizontal plane. The patient pulls the cable by means of a tape held in her hand. The cable is circled in red.

**Figure 6 sensors-23-06158-f006:**
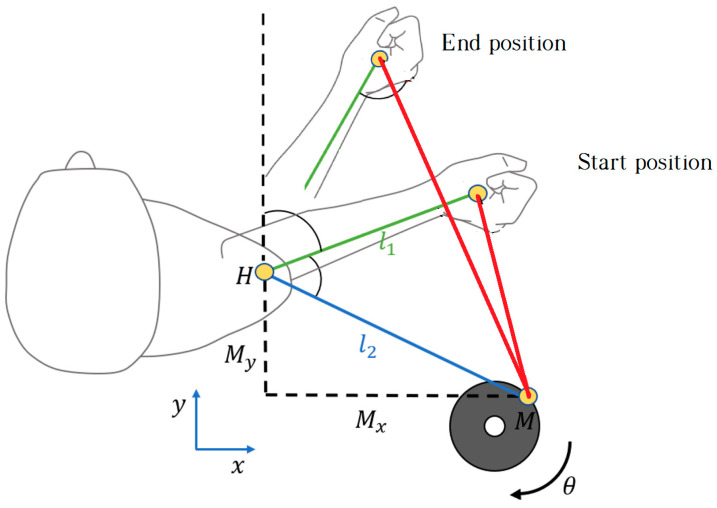
Range of movement for horizontal shoulder rotation. The rehabilitation session consists of the repetition of movements from the starting position to the end position. The starting position is defined by the physician and the end position is set according to the evolution of the patient.

**Figure 7 sensors-23-06158-f007:**
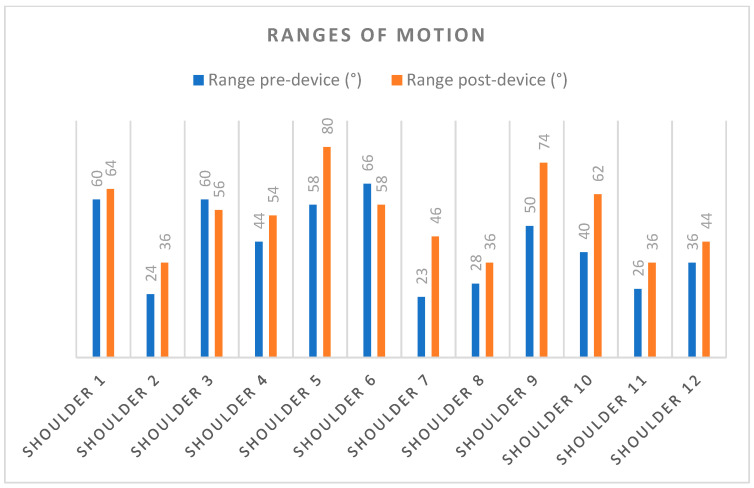
Ranges of motion after one rehabilitation session. A total of 10 out of 12 shoulders present improvements, while 2 present slight deteriorations, likely due to lack of familiarity with the device.

**Table 1 sensors-23-06158-t001:** Data of patients that tested the device. S11 and S12 were performed by the same patient, but with different shoulders (left and right, respectively).

Shoulder Session	Gender	Age	Pathology
1	M	66	Supraspinatus tendon rupture
2	F	45	Fracture with dislocation and calcified supraspinatus tendinosis
3	M	67	Complete rotator cuff rupture
4	M	57	Calcified supraspinatus tendinosis
5	M	46	Supraspinatus tendon suture due to subacromial syndrome
6	F	42	Supraspinatus tendinosis
7	M	66	Recovery after arthrolysis due to shoulder capsulitis
8	F	52	Supraspinatus tendon rupture
9	F	65	Rotator cuff tendinopathy
10	F	35	Supraspinatus tendinosis
11	F	61	Supraspinatus and infraspinatus rupture (left)
12	F	61	Tendinosis of the rotator cuff (right)

**Table 2 sensors-23-06158-t002:** Comparison of range of motion of shoulder external rotations.

Variable	Pre-Intervention Median (Interquartile Range)	Post-Intervention Median (Interquartile Range)	Intragroup Analysis Median *p*-Value
Range of motion Shoulder external rotations (degrees)	42.00 (33.00)	55.00 (25.50)	0.008 *

Data are expressed as the median and interquartile range. * *p* < 0.05 using the Wilcoxon test for related samples.

**Table 3 sensors-23-06158-t003:** Results of the system usability scale questionnaire.

Question	Score
Q1	4.75 ± 0.59
Q2	2.5 ± 1.55
Q3	4.66 ± 0.47
Q4	2.91 ± 1.49
Q5	4.08 ± 1.11
Q6	1.83 ± 1.28
Q7	4.58 ± 1.11
Q8	1.58 ± 1.32
Q9	4.91 ± 0.27
Q10	1.08 ± 0.27
TOTAL	80.71 ± 9.79

Data are expressed as mean ± standard deviation.

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
