# Peer review of "A Novel Active Device for Shoulder Rotation Based on Force Control"

_sensors, 2023, doi:10.3390/s23136158_

Round 1

Reviewer 1 Report

The paper discusses about prototyping a new device, called "Active device" which can be used for control rehabilitation, however,

1. Abstract does not contain background, which is the question addressed in a broad context and highlights the purpose of the study,

2. In the abstract, the author writes, "The results were excellent" seems not rational for the author to write this,

3. Introduction is not so clear, it does not define the purpose and significance.

4. Also, in the Introduction section, could not find what is the contribution of the research,

5. Could not find related work in the Introduction section,

6. In Section 2, there is not enough detail that should be included in the description of the materials so that the experiment can be reproduced,

7. In subsection 2.2, could not find any algorithm used in the research,

8. Could not find any methods used in the research,

9. In the research, also not clear how to evaluate,

5. In the results, the author did not compare the achieved results to the previous research.

Please check the language once more carefully. Really concerned about presentation and scientific soundness. Also, consider native speakers or professional help related to the research to ensure submission of the highest possible language level that authors really care about.

Author Response

the answers are attached in the file

Reviewer 2 Report

  The topic is very timely, needed, and hot. It covers a clear gap, and there is a lot of research going on currently in this area. Therefore, this paper deems an excellent future reference for researchers who work in the area. The overall quality is good and meeting the minimum requirements of the journal. The content is relevant and useful for the researchers' community. However, it has many minor issues that must be fixed properly ranging from literature and lack of suitable directions etc as follows:

1. The abstract must summarise the performance evaluation results and improvement over competitors and/or other solutions.

2. Please make sure that all keywords have been used in the abstract and the title.

3. The introduction section does not clearly describe the motivation of the research, please rewrite this section to make it clear.

4. The conclusions section should conclude that you have achieved from the study, contributions of the study to academics and practices, and recommendations of future works.

5. I suggest adding a brief description of each figure in its caption.

The paper needs rewriting and corrections. The language should be improved. Symbol naming and equations should be written with more care. Lower case and capital symbols should be unified.

Author Response

the answers are attached in the file.

Reviewer 3 Report

The authors present a good research paper. 

  • The relevance of the topic: Good.
  • Abstract: Can be improved.
  • Introduction: Can be improved.
  • Methodology: Can be improved.
  • Results: Good.
  • Discussion: Good.  
  • Conclusions: Can be improved.

However, ACCEPT AFTER MINOR REVISION. In general, the paper follows an adequate structure and correct scientific support and can be published considering some limitations. The study is interesting in the field of Shoulder injuries. However, there are a series of limitations that should be considered.

In the first place, carry out a review of the existing literature related to the subject, being essential to inquire into the MPDI – Sensors journal itself, since there are papers related to its manuscript that can help to improve it. Therefore, include those references, if any, especially from the last five years. In addition, recommend reading some papers related to the topic of Shoulder injuries:

Liaghat, B., Pedersen, J. R., Husted, R. S., Pedersen, L. L., Thorborg, K., & Juhl, C. B. (2023). Diagnosis, prevention and treatment of common shoulder injuries in sport: grading the evidence–a statement paper commissioned by the Danish Society of Sports Physical Therapy (DSSF). British Journal of Sports Medicine57(7), 408-416.

Enger, M., Skjaker, S. A., Nordsletten, L., Pripp, A. H., Melhuus, K., Moosmayer, S., & Brox, J. I. (2019). Sports-related acute shoulder injuries in an urban population. BMJ open sport & exercise medicine5(1), e000551.

Specific comments.

Title. The title of the manuscript is correct.

Abstract. Incorporate in the summary, a more precise sentence of the methodology and the results.

Introduction. This section presents the problem in a coherent and clear manner with the correct support of the scientific literature. However, it is convenient to update the references, since there are different documents related to the subject and no mention is made, and it would even be interesting to mention the different existing studies related to Shoulder injuries. Also, it could be a future study of review. Some bibliographical references are attached to carry out the section of Shoulder injuries:

Aydin, N., Kayaalp, M. E., Asansu, M., & Karaismailoglu, B. (2019). Treatment options for locked posterior shoulder dislocations and clinical outcomes. EFORT open reviews4(5), 194.

Methods. Add the Design section.

-       Study design. To write the design section, we recommend that you take some of the following methodologists as references.

Ato, M., López-García, J.J., & Benavente, A. (2013). A classification system for research designs in psychology. Annals of Psychology29(3), 1038-1059.

Results. Summary of study data and table are correct.

Discussion. The section Discussion is correct.

Conclusion. Differentiate the discussion of the main conclusions of the study. To do this, you must create this section. And modify the limitations of the study and locate them in said section at the end. Also, they must be direct, and highlight the main contributions of the study.

References. They should be reviewed and updated according to the publication standards. There are many errors in the references. Therefore, correct them and adapt them to the magazine's regulations.

Author Response

the answers are attached in the file.

Reviewer 4 Report

Dear Authors,

Thanks!

Please:

Abstract:

Explicitly state the aim and objectives of your research.

+

Participants:

n=? + Mean and SD?

Explicitly state the results and conclusions of your research.

“The results were excelent (!!!!!????), given that the range of movement of the shoulder increased in most of the patients.”

“The results of the Usability Scale questionnaire were also promising (!!!!????).”

Introduction:

 I would strongly advise the authors of this paper to rewrite their introduction to produce a more contextualised introduction toward a clear purpose.

For instance:

Line 39-42The rationale of exercise therapy is to improve load tolerance and possibly structural 39 adaptation of the musculotendinous unit to restore function. In the early phase of reha- 40 bilitation, flexibility exercises are often initiated and incorporated into strengthening re- 41 gimes to facilitate improvements in mobility (please, insert study/references)

--------------Line 70:  Please, insert aim and objectives of your research.

Materials and Methods

2.1. Device

 “The rehabilitation device consists of the following components: the actuator, trans- 73 mission components, force sensor, handle and controller. The architecture used is shown 74 in the Fig. 1. The actuator is based on a DC motor; the Maxon 75 DCX22S+GPX26HP+ENX16EASY was selected for this application. Its rated operation is 76 at 38 VDC, 5.9 Nm, and 10.5 revolution per minute (RPM). This motor generates the re- 77 quired force applied to the patient. The controller calculates the reference value according 78 to the desired force and information provided by sensors, then the driver (H-bridge 79 max14870) set the power to the motor The controller is run on a Texas Instruments board, 80 specifically, the LAUNCHXL-F28379D model. This board contains the 32-bit dual-core 81 TMS320F2-8379D microcontroller, which runs at 200 MHz. The controller closes the force 82 control loop at 1 kHz.The transmission is made up of a cable and a pulley that transmit 83 the actuator torque to the patient, the ratio between motor and pulley is 100:1 millimeter 84 A force sensor is located at the end of the cable and is also linked to the handle. The force 85 sensor is implemented by a load cell manufactured by Futek, which is used for measuring 86 the force reflected by the motor. The sensor model used in this work is LSB205 connected 87 to the Analog Strain Gauge Signal Conditioner IAA100, it offers a load measurement 88 range from 0 to 11 Kg corresponding to 0 to 5 DC volts.” (please, insert study/references)

Line 103:Fig. 2. This figure shows the main components of the rehabilitation device: actuator (1), 103 transmission components (2), force sensor (3) and handle (4) and controller (5).”

-------

Please, insert:

Fig. 2. Main components of the rehabilitation device: actuator (1), 103 transmission components (2), force sensor (3) and handle (4) and controller (5).

2.2. Control algorithm design

“In this work a force controller is proposed, it is based on the cable tension exerted by the 107 patient and an adaptive speed PI controller. The force sensor provides a range of detection 108 from 0 N to 111 N, therefore the maximum configurable payload of the prototype is 11.32 109 Kg. The proportional gain Kp is modified as a function of the ratio between the force error 110 and the reference, the proposed values are shown in Figure 3. Whereas the integral gain 111 Ki was set with a value of 2. The Kp and Ki were tuning experimentally.” (please, insert study/references)

2.3. Controller performance

Some trials were performed in the laboratory to properly tunned the controllers. Fig- 118 ure 4 shows the performance of the force exerted by the motor when is applied a step to 119 the reference. As we can see, the information provided by the force sensor is oscillating 120 on the reference. This oscillation is due to the interaction with the person, the different 121 movement speed is directly transformed to an incremental variation of the reflected force. 122 This is the usual behavior of a force control loop that maintain a reflected force. .” (please, insert study/references)

Participants

n=?

Mean and SD?

2.5. Intervention

“The clinician adjusted the controller parameters for each patient according to the 140 predefined exercise, in this case, the shoulder rotation movement. These parameters de- 141 fine the shoulder rotation range and the stress of the reflected force on the patient, which 142 is in an interval of 5N to 15N. Additionally, it was verified that the pulley was in the same 143 plane as the patient's forearm. This alignment makes movements more comfortable as the 144 forces directly oppose shoulder rotation with no movement at the elbow. 145 Each session lasted 15 minutes and consisted of repeating a cycle of external rotation 146 and internal rotation 30 times, returning to the starting position. These cycles were carried 147 out with the patient sitting down, in a chair with a backrest and with their feet on the 148 floor. Initially, the arm was positioned close to the body with a 90° elbow flexion, and 149 from this position, the rotation movements were performed. Figure 5 shows a patient dur- 150 ing a session”. (please, insert study/references)

To carry out the rotation movements, the patient started the external rotation movement 155 from the initial position up to the maximum rotation they were able to achieve. In this half 156 cycle, the patient exerted a force greater than that generated by the motor. In the second 157 half of the cycle, when the patient returned to the initial position, performing an internal 158 rotation, the movement was in the opposite direction and the force applied by the motor 159 was greater than that exerted by the patient. Figure 6 shows the limits of the cyclical move- 160 ment. The force reflected by the actuator was fixed throughout the cycle and was meas- 161 ured by the load cell. This load cell was located close to the patient's hand to obtain an 162 accurate estimate of the force reflected to the patient; otherwise, the forces caused by fric- 163 tion might affect the information processed by the controller.(please, insert study/references)

3.2. Clinical tests

The sample consisted of 12 shoulder rehabilitation sessions with 11 patients”

Ok, So:

Please:

Abstract (insert n=11, insert Mean and SD = xxxx);

Participants (n=11, insert Mean and SD = xxxx).

4. Discussion

“ The present study shows (????) the use of an active device for shoulder rehabilitation based  on a force control architecture. It aimed to rehabilitate, in a single treatment session, external rotation movements of the shoulder in patients in which this was limited by musculoskeletal injuries. It also describes the patients’ acceptance of the device, and results show a great (???) approval by patients after the first trial.”

Explicitly state the aim of your research.

However, this device is in line with a new generation of compact and simple rehabil- 298 itation equipment that are portable and easy for use by the patient [23], in contrast with 299 the classical robotic solutions previously mentioned. The main advantage of this kind of 300 solutions is the simplicity and cost-effective solution. Only one degree of freedom is 301 enough for performing the shoulder rehabilitation, and this equipment can be adapted for 302 horizontal and vertical rotations. Moreover, the reduce size of the device allows transport- 303 ing it to the rehabilitation workspace, that can be located in the clinic, or at patient home. 304 This prototype is currently improved in order to provide more functionalities to cli- 305 nicians for performing new rehabilitation exercises, and also developing software for a 306 better characterization of the rehabilitation sessions.” (---------------------------Study limitation?)

Conclusion

Please, insert pratical implications

References

The references must follow the guidelines.

For instance:

(…)

Kadivar Z, Beck CE, Rovekamp RN, O'Malley MK. Single limb cable driven wearable robotic device for upper extremity move- 380 ment support after traumatic brain injury. J Rehabil Assist Technol Eng. 2021. DOI: 10.1177/20556683211002448.

Gupta S, Agrawal A, Singla E. Architectural design and development of an upper-limb rehabilitation device: a modular syn- 382 thesis approach. Disabil Rehabil Assist Technol. 2022. DOI: 10.1080/17483107.2022.2071486.

(…)

Thank you for considering my suggestions, and I look forward to seeing your revised manuscript.

Sincerely,

Referee

-

Author Response

the answers are attached in the file.

Round 2

Reviewer 4 Report

Dear Authors,

Thanks!

Kind Regards 

-